# DISSECTING REGION-BASED REPRESENTATIONS

## ABSTRACT

Multimodal Large Language Models (MLLMs) typically process visual information as a flat sequence of image patch tokens, which is computationally expensive and lacks explicit semantic structure. This paper provides a systematic, vision-centric analysis of region-based representations, which group patches into semantically meaningful regions, as a more efficient and interpretable alternative. Our investigation is grounded in a key finding: MLLM performance is surprisingly robust to the input order of patch tokens, as the visual encoder already encode spatial information within the patches. This insight provides a foundational justification for reorganizing patches into semantically coherent regions. We further identify that the success of region-based methods depends on the quality of the visual features, particularly their smoothness and locality. We systematically evaluate how to enhance these properties through vision backbone selection, feature normalization, and hybrid partitioning strategies. Through comprehensive evaluations, we demonstrate that optimized region-based representations are a competitive alternative to patch-based ones, offering a compelling path towards more efficient, interpretable, and performant MLLMs.

## 1 INTRODUCTION

Multimodal Large Language Models (MLLMs) (Liu et al., 2023a; 2024a; Zhu et al., 2024; Dai et al., 2023) have achieved sophisticated capabilities in understanding and generating content across visual and textual domains. The dominant paradigm for their **visual encoding process** involves converting an image into a sequence of patch-based tokens using a Vision Transformer (ViT) (Dosovitskiy et al., 2021), which are then fed into the LLM via a simple projector (Liu et al., 2023a). While effective, this approach is computationally demanding due to the large number of visual tokens and lacks explicit semantic structure, often treating foreground objects and background clutter with equal importance.

An emerging and promising alternative is the **region-based representations** (Shlapentokh-Rothman et al., 2024), which group patch-based visual tokens into a smaller set of semantically meaningful regions before fusing them with the LLM. This approach holds the promise of significant gains in efficiency by reducing the number of visual tokens, and in interpretability by aligning tokens with recognizable image regions. However, despite their potential, a deep, systematic understanding of *why* region-based representations work and *how* to optimize their design for MLLMs remains largely under-explored. This work addresses this gap by providing a systematic, vision-centric analysis of region-based representations and their design principles.

Our investigation begins with a fundamental question: *why is reorganizing and aggregating patches into regions a viable and effective strategy?* A key insight from our work is that the performance of standard MLLMs is surprisingly robust to the relative ordering of patch-based tokens in the LLM sequence. In fact, ViT-based encoders (Dosovitskiy et al., 2021) already encode rich spatial information directly into the feature representation of each patch token. This token-order robustness provides a crucial justification for region-based approaches: if the strict grid order is not required for the LLM, then *reordering and reorganizing patches* based on representation similarity is a principled and effective strategy for creating more compact and structured visual inputs for MLLMs.

Building on this insight, we find that the primary factor leading to the success of region-based representations is the visual feature quality, specifically the *smoothness and locality* of the underlying visual features. Noisy or non-contiguous features can lead to poor region partitioning and aggregation, undermining the final performance. Through a series of controlled experiments, we identify several

key strategies to enhance region-based representations: 1) selecting pretrained vision backbones that produce smoother feature maps (*e.g.*, SigLIP (Zhai et al., 2023) or RADIO (Ranzinger et al., 2024b) over standard CLIP (Radford et al., 2021)); 2) applying additional feature normalization layers (Zhang & Sennrich, 2019) before aggregation; and 3) combining semantic-based partitioning from models like SAM (Kirillov et al., 2023) with feature-based clustering (Ester et al., 1996) to create more robust region-based representations.

To ground our analysis, we conduct comprehensive evaluations across a suite of MLLM benchmarks, including MMStar (Chen et al., 2024b), POPE (Li et al., 2023b), CV-Bench (Tong et al., 2024a), and OCR-Bench (Liu et al., 2023b). Our experiments demonstrate that well-designed region-based representations are a competitive alternative to the traditional patch-based counterpart, performing particularly well in vision-language tasks requiring *strong object and spatial awareness*. Beyond task performance, our analysis critically examines *efficiency and interpretability*. We demonstrate the token reduction benefits of region-based methods and use attention visualization to understand how these structured representations reveal the model's focus. Our main contributions are: (1) We provide a foundational insight justifying region-based representations by demonstrating that MLLM performance is robust to the input order of patch tokens, as spatial information is already encoded in the features. (2) We identify visual feature smoothness as a key factor for effective region-based representations and propose concrete strategies to improve it, including vision backbone selection, feature normalization, and hybrid partitioning methods. (3) We demonstrate empirically that optimized region-based representations are a competitive alternative to patch-based systems, offering a compelling path towards more efficient, interpretable, and performant MLLMs, especially on tasks requiring object-level understanding.

## 2 FROM PATCH TO REGION: ESTABLISHING A NEW PERCEPTION LEVEL

**Why Move Beyond Patches: Towards Vision-Centric Perception.** Modern MLLMs (Liu et al., 2023a; 2024a; Zhu et al., 2024; Dai et al., 2023) predominantly use a patch-based visual encoding paradigm. In this approach, a visual encoder divides an image into a fixed grid of patches, extracting a feature vector for each. These features are then projected by a connector module into the LLM's embedding space, creating a sequence of "visual tokens." These are concatenated with text prompts and fed into the LLM, which is then fine-tuned on visual instruction datasets to align the modalities. However, this standard patch-based approach has significant limitations. First, it scales poorly with image resolution: the number of visual tokens grows quadratically, leading to prohibitive computational costs and straining the LLM's context length. Second, a uniform grid is content-agnostic, arbitrarily dissecting objects and forcing the LLM to reconstruct concepts from a disjointed, low-level mosaic.

We argue for moving to region-based representations, where each visual token corresponds to a meaningful object or segment in the image. This shift offers a threefold advantage: 1) **Efficiency**: It breaks the quadratic scaling bottleneck, enabling high-resolution understanding with a relatively stable number of tokens. 2) **Semantic Grounding**: It provides the LLM with inputs that align with human-like perception, allowing for more direct reasoning about entities and their relationships. 3) **Interpretability**: The model's attention over meaningful regions can offer clearer insights into its reasoning process. These advantages frame region-based representations as a critical step toward building more scalable and vision-centric MLLMs.

**Basic Formulation of Region-based Representations.** The construction of region-based representations can be conceptualized as a two-stage process that operates on the output of a standard visual encoder: **(1) region partitioning** and **(2) feature aggregation**. First, given an input image, a pretrained visual encoder produces a grid of patch features, $F \in \mathbb{R}^{H \times W \times D}$, where $H \times W$ is the spatial resolution of the feature map and $D$ is the feature dimension. In parallel, a set of $K$ binary masks, $\{m_1, m_2, \ldots, m_K\}$, is generated, where each mask $m_k \in \{0, 1\}^{H \times W}$ defines a specific region by identifying the spatial locations of the patch features belonging to it. One way of deriving these masks is using open-world segmenters like SAM (Kirillov et al., 2023; Ravi et al., 2024), and then resizing the segmentation masks into the feature resolution. Alternative sources of regions, like clustering, will be discussed in the following sections.

In the second stage, feature aggregation, a single representative feature vector $r_k$ is computed for each region $k$ to replace the raw patch features. This is achieved by first selecting the subset of patch

features $F_k$ corresponding to the mask $m_k$, and then applying an aggregation function $\mathcal{A}$:

$$r_k = \mathcal{A}(\{F_{i,j} \mid m_k(i,j) = 1\})$$

The most straightforward aggregation method is to apply simple **average pooling**, where all patch features within a region are averaged to produce the final region feature. In the following sections, we will also discuss a more complex, cross-attention-based feature aggregation method.

Under this formulation, the conventional patch-based approach can be viewed as a special case of region-based representations where each patch constitutes its own singleton region. This perspective allows us to frame our investigation not as a comparison of two disparate methods, but as an analysis of the effects of moving along the perceptual level from a fine-grained, uniform grid to coarse-grained but semantically meaningful regions.

## 3 UNDERSTANDING REGION-BASED REPRESENTATION

Having established the motivation for region-based representations, we now turn to a systematic analysis of these concepts. In this section, we start by evaluating region-based representation, followed by additional observations on the rationale as well as the main challenge for region-based representations, as well as the potential remedies to mitigate the challenge. Our goal is not only to measure performance on standard benchmarks but also to gain a deeper understanding of the trade-offs between patch-based and various region-based approaches, especially concerning their efficiency and interpretability.

We followed the same training pipeline of LLaVA v1.5 (Liu et al., 2024a), using the same data and training hyperparameters. In all experiments, we also use the same LLM vicuna-7b-v1.5 (Zheng et al., 2023) to align with LLaVA v1.5. We use sam2.1-hiera-large (Ravi et al., 2024) as the segmentation model for generating regions. We tested over three visual encoders, CLIP (clip-vit-large-patch14-336) (Radford et al., 2021), SigLIP2 (google/siglip2-so400m-patch14-384) (Tschannen et al., 2025), RADIOv2.5 (radio-v2.5-l) (Heinrich et al., 2025).

### 3.1 EVALUATION ASPECTS

To ensure a thorough comparison, we extend our focus beyond benchmark performance to cover three critical aspects:

**Performance.** We evaluate the models' capabilities across seven benchmarks covering varying capabilities. POPE (Li et al., 2023b) focuses on measuring object hallucination by asking about whether specific objects are present in an image. OCRBench (Liu et al., 2023b) assesses the model's ability to read and comprehend text in images under various scenarios. CV-Bench (Tong et al., 2024a) repurposes classic vision tasks like spatial relationship and object counting to evaluate the model's spatial reasoning and fundamental 2D/3D perception. Finally, four comprehensive benchmarks MME (Fu et al., 2023), MM-Vet (Yu et al., 2024), MMBench (Liu et al., 2024b), and MMStar (Chen et al., 2024b), cover the diverse abilities of MLLMs, ranging from foundational perception tests to expert-level reasoning challenges. For CV-Bench, we slightly modified the prompt to align the prompt templates with other benchmarks. To facilitate a more detailed comparison, we further dive into specific sub-categories to reveal the specific advantages of region-based representation.

**Interpretability.** We assess the model's interpretability both qualitatively and quantitatively. Our analysis primarily focuses on the attention patterns the LLM assigns to visual tokens, as this provides insight into which parts of the image the model deems most important for its reasoning process. Additionally, we utilizes the annotations provided by PixCV-Bench (Siam, 2025), which contains segmentation masks of the object of interest for questions in CV-Bench, to compute a focus metric for quantitatively evaluating the attention attended to visual tokens during MLLM inference. Specifically, for each regular visual token, if the patch/region it represents has over a certain threshold of the area overlapping with the mask annotation, this token would be considered as a target token. Then, the focus metric is defined by the averaged total attention score of the answer tokens attending to target visual tokens.

**Efficiency.** We analyze the average number of visual tokens produced by different methods at various resolutions, which is particularly important for evaluating the efficiency gains of region-based

| Vision Encoder | #Tokens | POPE | OCRBench | CV-Bench | MMStar | MME Perception | MME Cognition | MMBench | MM-Vet |
|---|---|---|---|---|---|---|---|---|---|
| *Patch-based Representations* | | | | | | | | | |
| CLIP | 576 | 86.05 | 331 | 55.82 | 33.47 | 1499.80 | 300.00 | 66.84 | 32.1 |
| SigLIP2 | 729 | 85.98 | 405 | 60.03 | 35.27 | 1462.61 | 320.71 | 67.53 | 35.1 |
| RADIOv2.5 | 576 | 85.82 | 315 | 56.92 | 36.00 | 1494.28 | 304.64 | 67.35 | 29.60 |
| *Region-based Representations (from clustering)* | | | | | | | | | |
| CLIP | 257 | 85.80 | 310 | **57.66** | **35.80** | 1493.47 | **305.71** | 66.84 | 29.60 |
| SigLIP2 | 334 | 85.67 | 379 | 59.68 | **36.33** | **1542.31** | **323.57** | **69.67** | **35.8** |
| RADIOv2.5 | 124 | 84.51 | 280 | **58.10** | 35.47 | 1456.17 | **350.00** | 65.38 | 26.70 |

Table 1: MLLM performance under various vision encoders. We use the default resolution for CLIP and SigLIP2 and use $384$ for RADIOv2.5. #Tokens denotes the average visual token count per image. We report: F1 score for POPE, Perception & Cognition score for MME, and total or averaged score for all other benchmarks. Improved metrics from patch-based to region-based are **bolded**.

| Region Source | Resolution | #Tokens | Focus | OCRBench | MME OCR | MME code reasoning |
|---|---|---|---|---|---|---|
| Patch | | 576 | 10.57 | 315 | 125 | 42.5 |
| Segment | 384 | 101 | 16.22 | 250 | 125 | 57.5 |
| Cluster | | 124 | 13.39 | 280 | 125 | 50 |
| Combined | | 134 | 14.88 | 264 | 132.5 | 52.5 |
| Patch | 576 | 1296 | 10.91 | 357 | - | - |
| Segment | 768 | 104 | 15.80 | 275 | - | - |
| Combined | 768 | 159 | 14.72 | 260 | - | - |

Table 2: Visual token count, attention focus metric, as well as OCR performance under different resolutions and from different mask sources for region-based representations.

models and understanding the trade-offs between computational cost and performance. Similar to Interpretability, we also compute the average number of visual tokens produced by different region-based settings on CV-Bench and report it as a quantitative efficiency metric.

## 3.2 EVALUATING REGION-BASED REPRESENTATION

Table 1 shows our three evaluation aspects of different visual encoders under both patch and region-based settings. **Performance.** As we can see, all visual encoders demonstrate region-based representation as a competitive alternative to patch-based representations, with specific MME Cognition tasks showing consistent benefits. Breakdown performance on OCR-related tasks in Table 2 reveals that region-based representation does not always help OCR, which we hypothesize to be related to region mask quality. If each character is assigned an independent region, region-based representation should be helpful as it effectively separates characters. On the other hand, if multiple characters are crowded within the same region or some characters are not recognized as any region, region-based representation would conversely hurt, as it confuses or even loses information about certain characters. Our hypothesis is empirically validated by the visualizations of regions shown in Figure 2, where the quality of the segmented region matches the performance fluctuation. **Efficiency.** In Table 1, 2, all region-based representation variants show reduced visual token count, while RADIOv2.5 enjoys the most efficiency improvements. The rationale behind this will be discussed in Section 4.2. **Interpretability.** When compared with patch-based representation, all region-based representations provide more interpretable attention, characterized by the improved focus metric and attention visualizations shown in Fig. 1.

Summarizing the three aspects, we conclude that region-based representations work as a promising alternative for patch-based representations, and thus obtained the key finding 1:

**Finding 1**

**Region-based representations are competitive alternatives** to traditional patch-based representations. This is supported by competitive (or sometimes even improved) performance, improved efficiency, and better interpretability from our analysis.

## 3.3 NON-SENSITIVITY OF VISUAL TOKEN ORDERING

One potential concern about region-based representation might be that it breaks the predefined order of how LLMs receive visual tokens. In the patch-based representation, LLM receives all patch

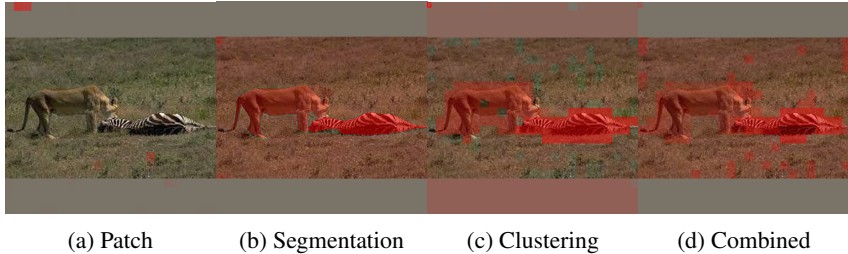

|          |               |            |             |
|:--------:|:-------------:|:----------:|:-----------:|
| (a) Patch | (b) Segmentation | (c) Clustering | (d) Combined |

Figure 1: Visualized attention to different parts of the image from different methods. Region-based methods all represent high interest in the zebra, while patch-based models only have high attention on either random or fixed positions. For region-based representations, we additionally visualize the attention of the global mask as a "G" letter top-left.

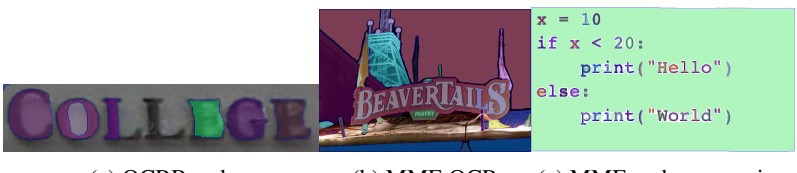

|          |               |            |
|:--------:|:-------------:|:----------:|
| (a) OCRBench | (b) MME OCR | (c) MME code reeasoning |

Figure 2: Visualized regions from segmentation, for OCRBench and MME code reasoning subset.

| Setting | POPE | OCRBench | CV-Bench | MMStar |
|---|---|---|---|---|
| *Patch-based* | | | | |
| $\mathcal{A}$: CLIP+RMSNorm | 85.83 | 317 | 58.19 | 33.93 |
| $\mathcal{B}$: $\mathcal{A}$+random order | 85.73 | 312 | 58.80 | 34.27 |
| $\mathcal{C}$: RADIOv2.5+RMSNorm | 85.43 | 319 | 58.54 | 34.60 |
| $\mathcal{D}$: $\mathcal{C}$+random order(trained) | 85.10 | 313 | 57.45 | 36.33 |
| $\mathcal{E}$: $\mathcal{C}$+random order(w/o/ training) | - | - | - | 35.07 |
| $\mathcal{F}$: $\mathcal{C}$+pre-shuffle | - | - | - | 28.27 |
| $\mathcal{G}$: $\mathcal{D}$+pre-shuffle | - | - | - | 28.20 |
| *Region-based (from combined source, at 768x resolution)* | | | | |
| $\mathcal{H}$: RADIOv2.5 | 86.43 | 275 | 57.15 | 33.93 |
| $\mathcal{I}$: $\mathcal{H}$+random order | 86.43 | 279 | 57.98 | 32.80 |

Table 3: Effect of the order of visual tokens. `pre-shuffle` means randomly shuffle the image patches before the visual encoder, rather than randomly shuffle the visual tokens fed into LLM.

representations in a default scanline order, which is intuitive and follows fixed patterns, which might be a soft-encoding of the spatial information in the image. However, regions in an image can have an irregular shape, while the "correct" order for those regions remains unclear. To address this, we sort the regions according to their center-of-mass by default, and compare the scanline order with a random order under both patch-based and region-based settings in Table **??**. From the results, the order of the visual tokens has no or a negligible impact on preserving MLLM's spatial relations, regardless of whether MLLM is trained to adapt such order. On the other hand, shuffling image patches would immediately degrade performance, indicating that the spatial information are primarily encoded in the positional embeddings in the visual encoder.

> **Finding 2**
>
> The spatial information of visual tokens is **encoded in the learned positional embeddings in the visual encoder**, rather than the visual tokens' relative order.

## 4 MAJOR CHALLENGE FOR REGION-BASED REPRESENTATIONS

While the concept of aggregating patch features into region-level representations is intuitive and competitive, its effectiveness is fundamentally contingent on the quality and consistency of the

underlying patch features. In this section, we propose a critical and often overlooked challenge from the visual encoder aspect for region-based representations, followed by approaches that can mitigate this issue.

## 4.1 VISUAL FEATURE INCOHERENCE IN VISUAL ENCODERS

The fundamental premise of feature aggregation is that patch features within the same region can be effectively compressed into a region-level feature with minimal information loss, which naturally relies on the following assumption: Raw features from visual encoders are **spatially coherent**. That is, 1) the features primarily contain information about the local patch, and 2) the features of two adjacent and semantically similar patches are also similar.

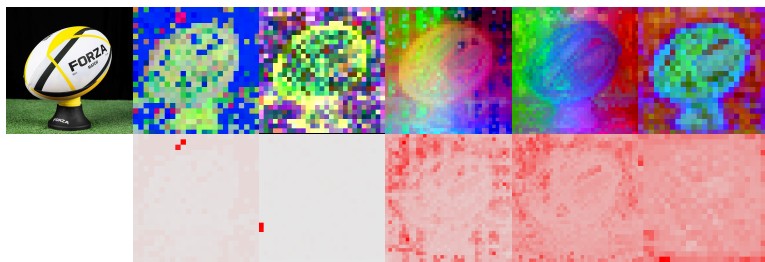

Figure 3: Visualized raw features and corresponding feature norms. Features are visualized through principal component analysis (PCA). CLIP and SigLIP2 show high-norm artifacts known as "registers", while other models produce smoother features and less extreme high-norm artifacts.

However, the above assumption is frequently violated by modern visual encoders, particularly those supervised by image-text pairs such as the CLIP series (Radford et al., 2021; Cherti et al., 2023; Fang et al., 2024) and the SigLIP series (Zhai et al., 2023; Tschannen et al., 2025). On the other hand, visual encoders like DINOv2 (Oquab et al., 2023) have better feature coherence due to their self-supervised training objective, but they are never pre-aligned with the language modality, making them unsuitable for MLLM integration. Figure 3 demonstrates the feature incoherency of common visual encoders.

**Decomposing Incoherence.** Our observation of visual feature incoherence manifests two primary factors: **high-norm artifacts** and **non-smoothness**.

One well-documented source of incoherence is the presence of **high-norm artifacts** in the feature map. It has been shown that Vision Transformers (ViTs) spontaneously learn to use semantically sparse patches as "registers" to store global information during processing, and explicitly introducing a small number of register tokens helps mitigate this phenomenon (Darcet et al., 2024). These register patches develop feature vectors with exceptionally high norms that are unrepresentative of their local visual content. When these artifact features are included in the feature aggregation for a larger region, their high magnitude can dominate the resulting representations, effectively corrupting the aggregated feature and diverting the LLM's attention. Figure 3 demonstrates these "register" artifacts in the feature norm map.

Beyond these distinct artifacts, we also observe a more general **non-smoothness** in the feature space. Ideally, adjacent patches that belong to the same object or surface should have highly similar feature vectors, creating a smooth transition across the feature map. Yet, many language-supervised encoders produce features with surprisingly low similarity between neighboring patches. This noisy, disjointed representation makes it difficult to learn localized information. When aggregating features across a semantic region, this non-smoothness causes averaging a collection of dissimilar vectors, washing out subtle but important details, and eventually failing to produce a truly representative feature for the region. Denoising-ViT (Yang et al., 2024) considers such non-smoothness as noise in the feature map, and further shows that these artifacts can be traced back to the learned patterns in the positional embeddings, and can be removed by appending a denoising layer in the end.

> **Finding 3**
>
> The **incoherency** of the raw visual features is the major challenge in the development of region-based representation, characterized by **high-norm artifacts** as well as **non-smoothness** between adjacent patches.

## 4.2 ATTEMPTS TO MITIGATE INCOHERENCE

Given that visual feature incoherence is a fundamental challenge, we explore several distinct but complementary strategies to mitigate its negative impact on region-based representations. These approaches operate at different stages of the visual encoding procedure: (1) seeking more coherent raw features; (2) processing the raw features through normalization; (3) revisiting different sources of obtaining region masks; and (4) alternative, learnable feature aggregation approaches.

**Agglomerative Visual Encoders**   Recent works (Tong et al., 2024a; Shi et al., 2025; Li et al., 2025c;a) have explored utilizing visual representations from multiple visual encoders or unfreezing the visual encoders to further boost the performance. Despite their efforts might mitigate the incoherency issue to some extent, these approaches might introduce extra complexities and does not help when a single visual encoder is enforced. To better compare different visual encoders and simplify training and evaluation, we focus on using a single frozen visual encoder.

In addition to adopting multiple visual encoders at the same time, agglomerative visual encoders (Heinrich et al., 2025; Sariyildiz et al., 2024; Shang et al., 2024) combine the strengths of different visual encoders through multi-teacher agglomeration. By jointly distilling from CLIP (Radford et al., 2021), SigLIP (Zhai et al., 2023), DINOv2 (Oquab et al., 2023), and SAM (Kirillov et al., 2023), these models obtain both language alignment abilities from language-supervised models as well as fine-grained perception and segmentation abilities from self-supervised models. In terms of the feature coherence, RADIOv2.5 (Heinrich et al., 2025) reaches a better balance between language alignment and feature coherence, as shown in Figure 3. Reduced token counts in Table 1 also confirm the effectiveness of adopting RADIOv2.5, where improved feature coherency allows RADIOv2.5 to create fewer regions through clustering introduced in Sec 4.2.

**Normalization**   The high-norm artifacts present in visual features are particularly problematic even for patch-based MLLMs, as these outlier tokens can disproportionately capture the model's attention, rendering other informative visual tokens ineffective. A direct and efficient way to address this is to apply a normalization layer immediately after extracting features from the visual encoder. This simple addition helps to tame the magnitude of outlier features, preventing them from dominating the subsequent attention mechanisms. It's worth noting that this step may be less critical for RADIOv2.5, as it already applies PHI-S (Ranzinger et al., 2024a) to normalize teacher representations in its training objective, resulting in student features that are inherently more uniform in magnitude.

Table 4 compares the results with/without RMSNorm (Zhang & Sennrich, 2019) normalization on the raw features of visual encoders. Overall, as shown by the MMStar performance changes, region-based representations benefit from normalization, especially on the comprehensive benchmark MMStar. Patch-based representations, though, also benefit from removing high-norm outliers and derive more localized attention patterns as shown in Figure 6 in the appendix, but do not receive performance gains. We attribute this to the information loss from the rescaling behavior of normalization.

**Different sources of region**   Here, we investigate three primary sources for deriving regions, each offering a different trade-off between semantic grounding and feature consistency.

From Segmentation. Generating masks from an open-world segmenter like SAM (Kirillov et al., 2023; Ravi et al., 2024) is the most straightforward way of obtaining regions. This approach prioritizes semantics, as the resulting regions directly correspond to objects or background of the scene. In this work, we mainly adopt the design proposed by Shlapentokh-Rothman et al. (2024), but remove the SLIC (Achanta et al., 2012) refinement step for simplicity.

While this method ensures that regions are semantically meaningful, it is agnostic to the underlying patch features. A single segmentation mask can easily encompass a set of highly incoherent or noisy

| Vision Encoder | POPE | OCRBench | CV-Bench | MMStar |
|---|---|---|---|---|
| *No Normalization* | | | | |
| CLIP | 86.05 | 331 | 55.82 | 33.47 |
| SigLIP2 | 85.98 | 405 | 60.03 | 35.27 |
| RADIO | 85.82 | 315 | 56.92 | 36.00 |
| *Using RMSNorm* | | | | |
| CLIP | 85.39 | 321 | 56.09 | 32.87 |
| SigLIP2 | 86.17 | 389 | 57.67 | 34.93 |
| RADIO | 85.26 | 314 | 57.67 | 34.80 |
| *Region-based Representations (from clustering)* | | | | |
| **Normalization** | POPE | OCRBench | CV-Bench | MMStar |
| No | 84.91 | 273 | 58.31 | 33.13 |
| RMSNorm | 84.51 | 280 | 58.10 | 35.47 |

Table 4: With and without normalization for patch-based models. Upper: results for patch-based representations; Below: results for region-based representations (from clustering).

patch features, especially when using CLIP-like visual encoders. Aggregating these features, for instance through simple averaging, can lead to a representation that is not truly representative of the region's visual content.

From Clustering. An alternative approach is to generate regions by directly clustering the patch features themselves with respect to spatial locality as well as feature similarity. This method explicitly addresses the feature coherence challenge by grouping patch features based on similarity. This idea coincides with ToMe (Bolya et al., 2023; Bolya & Hoffman, 2023), which is later adopted in MLLM token compression works (Weng et al., 2024; Li et al., 2024), but here we still view this as a method for generating region masks. Compared with the standard ToMe, our implementation includes two key modifications: (1) we apply token merging only after the final layer of the visual encoder; and (2) we set a threshold on feature similarity instead of merging to a fixed number of regions.

By design, this approach produces regions with high internal feature consistency, but comes with a trade-off: the shape of a cluster-based region can be arbitrary and may not align with any clear semantic concept. As shown in Figure 4, this method may also produce small, single-patch clusters that correspond to feature artifacts.

Combining Segmentation and Clustering. To get the best of both worlds, we additionally propose a hybrid method that combines the semantic grounding of segmentation with the feature-coherent properties of clustering. The process is as follows: first, we use SAM to generate an initial set of semantically coherent regions. Then, for any large regions generated by segmentation, we apply a clustering algorithm to further split them based on patch feature similarity. Since we are already operating within a semantically meaningful area, we no longer need to consider the spatial localities of the patches during clustering. We then apply the classic clustering algorithm DBSCAN (Ester et al., 1996) to partition patch features. This combined approach aims to balance both semantic and feature consistency, producing regions that are more suitable for robust feature aggregation. Figure 4 provides a visualization of regions derived from all three sources, illustrating how the combined method resembles a mixture of the two.

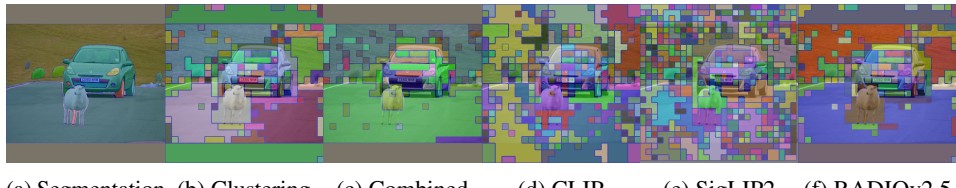

(a) Segmentation  (b) Clustering  (c) Combined  (d) CLIP  (e) SigLIP2  (f) RADIOv2.5

Figure 4: Left: Regions from different sources. Patch features are derived from RADIOv2.5 (Heinrich et al., 2025). right: Regions generated from clustering on different patch features.

| Region Source | POPE | OCRBench | CV-Bench | MMStar | | | | | | |
| | | | | CP | FP | IR | LR | ST | MA | Avg |
|---|---|---|---|---|---|---|---|---|---|---|
| *Average-pooling aggregation* | | | | | | | | | | |
| SAM | 86.09 | 255 | 56.41 | 58.40 | 32.80 | 40.00 | 30.40 | 19.60 | 29.20 | 35.07 |
| Combined | 86.55 | 277 | 56.73 | 56.80 | 34.00 | 42.40 | 30.00 | 24.80 | 25.20 | 35.53 |
| *Cross-attention aggregation* | | | | | | | | | | |
| SAM | 85.70 | 275 | 57.84 | 58.80 | 32.00 | 40.40 | 30.00 | 19.60 | 24.40 | 34.20 |
| Combined | 85.79 | 260 | 56.54 | 55.20 | 30.40 | 42.00 | 31.60 | 16.80 | 27.20 | 33.87 |

Table 5: Different region representation aggregation methods under 768x. MLLMs not always benefit from increasing resolution.

For the clustering-based region source, we fix the threshold to be $0.7$ using cosine similarity. For combined region source, we adopt the implementation of DBSCAN in scikit-learn (Pedregosa et al., 2011) and RAPIDS cuML (Raschka et al., 2020), and apply clustering on segmented regions with more than $10$ patches. Due to implementation issues, we use *eps=0.7, min_samples=3* under normalized L2 distance in practice, which is equivalent to cosine distance under another threshold. Figure 4 visualizes regions generated from different sources and patch features. Consistent with our previous conclusion and main results in Table 1, RADIOv2.5 produces less and more sementically meaningful clusters, demonstrating its advantage in adopting region-based representation.

**Cross-Attention Feature Aggregation** We have introduced simple average pooling as a feature aggregation method, where all patch features within a region are averaged to produce a single region-level feature. While simple, this approach may not be optimal, especially when dealing with the feature incoherence discussed above.

To address this issue, we propose an alternative, learnable aggregation approach using a multi-head cross-attention module, in which the pooled feature is added by a learnable bias and attends to the patch features belonging to that region. This allows the model to dynamically learn which patch features are more representative and up-scale their importance, while down-weighting the influence of noisy or outlier features. The resulting aggregated representations can potentially retain more selective and nuanced information, making it a more robust way to form region features despite the underlying feature incoherence.

However, the results in Table 5 do not show a meaningful advantage of cross-attention compared with simple pooling. We suspect a simple cross-attention module is still not expressive enough to handle the incoherence, and a more complex design might be needed to make a difference from simple pooling.

Based on the above analysis, we summarize the last finding:

**Finding 4**

Multiple potential solutions exist to mitigate incoherence. The most effective and fundamental way is to adopt a more coherent visual encoder, *e.g.* RADIOv2.5. Other solutions include normalizing features before aggregation, switching or combining different sources of regions, but at a cost of information loss and sacrificing the semantic consistency of regions. Adopting a more complex aggregation mechanism might also work, but a more complex design is needed.

## 5 CONCLUSION

In this work, we demonstrate that region-based visual representations are a compelling alternative to the conventional patch-based encoding in MLLMs. Our key insight that MLLMs are robust to the input order of patch tokens justifies the principled reorganization of patches into semantic regions. The success of region-based representations relies on smooth and localized visual features, and thus the performance can be enhanced through vision backbone selection, feature normalization, and hybrid region partitioning. Our findings provide an actionable framework for developing MLLMs that are more efficient, interpretable, and effective on tasks requiring object-level understanding.

## ETHICS STATEMENT

This work shares the potential ethical risks common to other Multimodal Large Language Models (MLLMs), including the capacity to generate harmful content or amplify existing societal biases. Our study is conducted exclusively using publicly available and open-source models, training data, and evaluation benchmarks, thereby avoiding the use of private or sensitive personal information. We acknowledge, however, that biases present in these foundational resources may be inherited and reflected in our experimental outcomes. While the focus of this paper is a technical analysis of visual encoding rather than direct fairness mitigation, we believe that a deeper understanding of model architecture is a crucial step toward developing more robust and equitable systems. We are committed to transparency, and all resources used are cited to support reproducibility and encourage community-driven scrutiny of these important ethical considerations.

## REPRODUCIBILITY STATEMENT

We are committed to ensuring the reproducibility of our research. Our entire experimental framework is built upon publicly available, open-source resources; the specific pretrained vision encoders, large language models, training data, and evaluation benchmarks used are detailed in the experimental setup section and the appendix. We provide comprehensive implementation details, including all hyperparameters and training configurations, in the supplementary material. To facilitate direct replication and extension of our work, we will release our complete source code upon the paper's acceptance. We believe these measures provide the community with all the necessary components to reproduce our results and build upon our analysis.

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

## A  RELATED WORK

**Multiple Large Language Models (MLLMs).** Building upon Large Language Models (LLMs), MLLMs integrates the ability to perceive and reason about visual information. Pioneering works such as LLaVA (Liu et al., 2023a; 2024a), MiniGPT-4 (Zhu et al., 2024; Chen et al., 2023a), and InstructBLIP (Dai et al., 2023; Li et al., 2023a) have established the fundamental architecture that connects a pre-trained visual encoder and a pre-trained LLM with an MLP-based or transformer-based connector via visual instruction tuning. Recent study on MLLMs is centered around enhancing MLLM capabilities in visual grounding (Peng et al., 2024; Chen et al., 2023b; Lai et al., 2024; Rasheed et al., 2024), complex reasoning (Dong et al., 2025; Guo et al., 2024), and unified generation (Chen et al., 2025; Fan et al., 2025), mainly from the data perspective. In contrast, this work examines the relatively less explored component of visual information.

**Visual Encoders for MLLMs.** Pre-trained with vision-language alignment, CLIP (Radford et al., 2021) is the predominant visual encoder in early MLLMs. Variants of CLIP (Zhai et al., 2023; Tschannen et al., 2025; Fang et al., 2024) present improved vision-language understanding and are widely adopted. Recent study (Tong et al., 2024b;a) reveals the inherent shortcomings in CLIP representation, and proposes to combine multiple visual encoders (*e.g.*, DINOv2 (Oquab et al., 2023) and convolutional CLIP (Liu et al., 2022; Cherti et al., 2023)) in MLLMs. Also, agglomerative models (Ranzinger et al., 2024b; Heinrich et al., 2025; Shang et al., 2024; Sariyildiz et al., 2024; Lu et al., 2025) combine the strengths of multiple teachers into one unified encoder. In this work, we compare choices of CLIP (Radford et al., 2021), SigLIP2 (Tschannen et al., 2025), and RADIOv2.5 (Heinrich et al., 2025) in detail.

**Region-Based Representations and Token Merging.** The idea of grouping image pixels into semantically coherent regions and perceiving them as structured elements is recently revisited (Shlapentokh-Rothman et al., 2024; Garg et al., 2024; Khosla et al., 2025b; Xiao et al., 2025; Khosla et al., 2025a) given the advancements in deep visual representations (Oquab et al., 2023; Radford et al., 2021) and the segment anything models (Kirillov et al., 2023; Ravi et al., 2024). Compared with patch tokens directly from ViTs (Dosovitskiy et al., 2021), region-based representations can significantly reduce the number of tokens to process. Meanwhile, token merging methods (Bolya & Hoffman, 2023; Shang et al., 2025; Chen et al., 2024a; Cai et al., 2025; Yang et al., 2025b; Li et al., 2025b) combine multiple patch tokens into fewer while more informative ones for MLLMs. We investigate the strengths of region-based token merging through comprehensive experiments.

## B  SUMMARY OF KEY FINDINGS (FROM MAIN PAPER)

Here, we reiterate the key findings in the main paper for better reference:

1. **Region-based representations are competitive alternatives** to traditional patch-based representations. This is supported by competitive (or sometimes even improved) performance, improved efficiency, and better interpretability from our analysis.
2. The spatial information of visual tokens is **encoded in the learned positional embeddings in the visual encoder**, rather than the visual tokens' relative order.
3. The **incoherency** of the raw visual features is the major challenge in the development of region-based representation, characterized by **high-norm artifacts** as well as **non-smoothness** between adjacent patches.
4. Multiple potential solutions exist to mitigate incoherence. The most effective and fundamental way is to adopt a more coherent visual encoder, *e.g.* RADIOv2.5. Other solutions include normalizing features before aggregation, switching or combining different sources of regions, but at a cost of information loss and sacrificing the semantic consistency of regions. Adopting a more complex aggregation mechanism might also work, but a more complex design is needed.

Fig. 5 demonstrates the approaches we investigate to mitigate incoherence.

## C  LIMITATIONS AND FUTURE WORK

In this work, we mainly focus on evaluating the design choices of encoding region-based visual information in MLLMs, and stick to a relatively simple MLLM training pipeline with a **frozen visual**

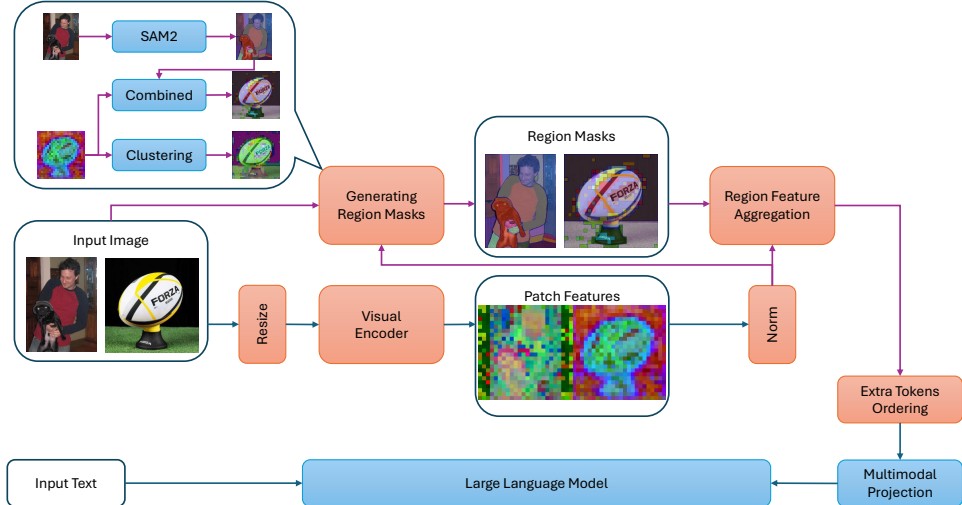

Figure 5: Approaches to mitigate incoherence. The orange boxes denote the design choices we investigate, while the blue boxes denote the fixed part.

**encoder** to better understand the native differences between visual encoders. For the same reason, we only conduct experiments on language-supervised visual encoders. Potential future steps may include switching to more recent training recipes and data, and unfreezing the visual encoders to observe how different visual encoders can benefit from further fine-tuning.

**Long video understanding** tasks are not included in our evaluation, because fitting in more frames under the same context length is a key issue Weng et al. (2024); Chen et al. (2024c). This might bring an additional key advantage of region-based representations, as they produce fewer visual tokens.

Despite comparing multiple sources of regions, our exploration in the region-based representation aggregation part does not bring uniform performance gains, leaving a potential direction for alternative designs that surpass simple pooling.

## D    BROADER IMPACTS

This work shares the common risks associated with other MLLMs, including the potential to introduce or amplify existing societal biases. While we rely solely on publicly available benchmarks, models, and training data, which avoids private or personal information, biases present in these resources may still influence final outcomes. We do not specifically target fairness mitigation in this study, but we recognize its importance and encourage future research to address these concerns. All resources used are publicly released to support transparency, reproducibility, and community-driven scrutiny.

## E    EXPERIMENT DETAILS

### E.1    IMPLEMENTATION DETAILS

**Generating Regions from SAM.** As mentioned in the main paper, we use SAM2's Ravi et al. (2024) automatic mask generator for deriving the segmentation of the whole image. In practice, we observe that the number of masks produced from each image largely depends on the contents of the image. To control the number of regions at a consistent level, we set multiple segmentation granularities with an increasing points-per-side parameter and decreasing mask filtering parameters. When increasing granularity, the automatic mask generator is provided with more point prompts and filter out fewer low-quality masks, therefore producing more masks in the end. We start at an

intermediate granularity, and re-generate segmentation masks at a higher/lower granularity when the number of masks produced exceeds the soft lower/upper bound.

Different from Shlapentokh-Rothman et al. (2024)'s default implementation of upsampling the features into the size of the masks, we alternatively downsample the masks into the size of the patch features to reduce both GPU memory usage and computation cost, as well as aligning with the setting of the clustering alternative. Specifically, masks are first preprocessed into the same size as the input resolution of the vision encoder (*e.g.*, $384 \times 384$, following similar steps including padding and resizing as the input image), and then downsampled to the size of the patch-level feature map (*e.g.*, $24 \times 24$). Since each image patch is reduced to one pixel in the patch-level feature map, we alternatively use the following downsampling method: we average pool masks within each image patch, and those pixels with pooled values higher than a certain threshold are retained in the mask. This means whether a downsampled pixel is retained depends on the size of the overlapping area between the mask and the original patch. By setting a small threshold, we can avoid small regions from vanishing unless they are extremely small.

**Generating Regions from Clustering.** We mainly adopt UnSAM's Wang et al. (2024) iterative merging as the clustering method:

1. We start by setting each patch as a single region;
2. A region's feature is defined by the average of the patch features covered by the region;
3. We iteratively merge adjacent regions with the highest feature cosine similarity, until it drops below a certain threshold;
4. The remaining regions after iterative merging are considered to be the result regions.

We additionally use priority queues and disjoint sets to optimize the clustering process.

## E.2 HYPERPARAMETERS

**Training.** We adopt the same training hyperparameters of LLaVA-1.5 Liu et al. (2024a): `total batch size=256, learning rate=1e-3` for visual feature alignment, `total batch size=128, learning rate=2e-5` for visual instruction tuning. For both stages, we train only one epoch with `bfloat16` precision using a fixed seed `42`.

**Inference.** During inference, we set `temperature=0`, or equivalently, greedy decoding on all benchmarks. For short answer tasks, we unify the question prompt to be *¡image¿ + [question] + "Answer the question using a single word or phrase."* For QA tasks, we unify the question prompt to be *¡image¿ + [question] + [options] + "Answer with the option's letter from the given choices directly."*

**Region-based Representation.** When generating regions using SAM, we set three segmentation granularities with the following parameters: `points-per-side=48,64,96`, `pred-iou-thresh=0.6,0.5,0.4`, `stability-score-thresh=0.92,0.9,0.85`, and set the soft bounds on the number of regions to be $[80, 160]$. For MMstar Chen et al. (2024b) and OCRBench Liu et al. (2023b), we only keep the first two granularities and reduce the soft lower bound to $64$ for simplicity and faster generation. When downsampling masks, we set the threshold of valid pixels to be $0.07$.

When using clustering to generate regions, we set the similarity threshold to be $0.7$. When combining segmentation and clustering, we use DBSCAN Ester et al. (1996) with `metric=l2, eps=0.7, min samples=3` to further split the segmentation masks containing at least 10 patches into smaller regions according to the normalized patch features.

In cross-attention region feature aggregation, we set `num heads=16`, and use the average-pooled patch feature as the single query token.

## E.3 COMPUTATIONAL RESOURCES

All training experiments are conducted on 4 NVIDIA H100 GPUs, which take roughly 16 hours to complete each two-stage training under the default setting for patch-based representations. Other settings have fluctuated training time from 7 hours to two days according to the specific setting. Inference is conducted on a single NVIDIA H100 GPU.

| Region Source | Res | Norm | Notes | #Tok | Focus | POPE | OCRBench | | | | | CV-Bench | | | |
|---|---|---|---|---|---|---|---|---|---|---|---|---|---|---|---|
| | | | | | | | TR | ST-C | D-O | KIE | HMER | Count | Relation | Distance | Depth |
| *CLIP* | | | | | | | | | | | | | | | |
| Patch | | no | | 576 | 12.54 | 86.05 | 173 | 128 | 19 | 11 | 0 | 48.73 | 59.08 | 68.00 | 50.17 |
| Patch | | rms | | 576 | 11.96 | 85.39 | 171 | 124 | 18 | 8 | 0 | 46.95 | 56.77 | 70.50 | 52.83 |
| Patch | | rms | -CLS | 576 | - | 85.83 | 168 | 124 | 19 | 6 | 0 | 51.27 | 63.38 | 69.17 | 51.67 |
| Patch | 336 | rms | -order | 576 | - | 85.74 | 170 | 120 | 18 | 4 | 0 | 48.48 | 64.46 | 72.67 | 52.83 |
| Segment | | rms | | 101 | 15.54 | 86.21 | 153 | 104 | 13 | 5 | 0 | 45.05 | 60.31 | 75.33 | 53.33 |
| Cluster | | rms | | 257 | 12.72 | 85.80 | 168 | 124 | 15 | 3 | 0 | 45.81 | 62.92 | 73.67 | 51.67 |
| Combined | | rms | | 139 | 14.41 | 85.48 | 157 | 113 | 14 | 4 | 0 | 46.07 | 59.54 | 73.00 | 53.17 |
| *SigLIP2* | | | | | | | | | | | | | | | |
| Patch | | no | | 729 | 11.31 | 85.99 | 217 | 146 | 22 | 20 | 0 | 56.35 | 69.85 | 65.67 | 50.83 |
| Patch | | rms | | 729 | 11.51 | 86.17 | 210 | 146 | 19 | 14 | 0 | 57.36 | 67.08 | 58.67 | 49.67 |
| Segment | 378 | rms | | 101 | 15.24 | 84.96 | 180 | 98 | 18 | 5 | 0 | 48.48 | 63.23 | 71.17 | 50.00 |
| Cluster | | rms | | 334 | 8.34 | 85.68 | 200 | 141 | 24 | 14 | 0 | 54.95 | 67.54 | 65.17 | 53.67 |
| Combined | | rms | | 142 | 14.21 | 85.40 | 192 | 117 | 20 | 6 | 0 | 51.90 | 61.69 | 74.00 | 53.50 |
| *RADIO* | | | | | | | | | | | | | | | |
| Patch | | no | | 576 | 10.57 | 85.82 | 189 | 106 | 11 | 9 | 0 | 52.16 | 62.62 | 64.83 | 50.83 |
| Patch | | rms | | 576 | 12.44 | 85.25 | 189 | 107 | 11 | 7 | 0 | 52.79 | 63.85 | 67.17 | 49.17 |
| Segment | | no | | 101 | 16.29 | 84.82 | 148 | 79 | 9 | 1 | 0 | 46.83 | 65.85 | 64.50 | 50.17 |
| Segment | 384 | rms | | 101 | 16.22 | 84.32 | 156 | 81 | 10 | 3 | 0 | 46.32 | 60.15 | 69.17 | 55.17 |
| Cluster | | no | | 117 | 13.50 | 84.91 | 154 | 103 | 12 | 4 | 0 | 51.14 | 64.15 | 68.33 | 52.67 |
| Cluster | | rms | | 124 | 13.39 | 84.51 | 160 | 102 | 13 | 5 | 0 | 49.24 | 60.62 | 71.50 | 53.67 |
| Combined | | no | | 134 | 14.86 | 84.66 | 157 | 89 | 9 | 5 | 0 | 50.76 | 64.00 | 68.67 | 52.17 |
| Combined | | rms | | 134 | 14.88 | 84.76 | 162 | 89 | 9 | 4 | 0 | 48.10 | 62.92 | 72.67 | 56.83 |
| Patch | 576 | no | | 1296 | 10.91 | 86.92 | 196 | 127 | 16 | 18 | 0 | 52.41 | 67.08 | 68.67 | 53.33 |
| Segment | 576 | rms | | 104 | 15.80 | 85.91 | 156 | 94 | 14 | 6 | 0 | 46.45 | 64.46 | 69.00 | 53.50 |
| Combined | 576 | rms | | 159 | 14.72 | 86.05 | 162 | 103 | 14 | 5 | 0 | 48.10 | 62.77 | 66.67 | 53.33 |
| Segment | 768 | no | | 105 | 15.67 | 85.70 | 165 | 97 | 9 | 4 | 0 | 51.40 | 61.08 | 68.83 | 52.50 |
| Combined | 768 | no | | 154 | 14.46 | 85.79 | 146 | 98 | 14 | 2 | 0 | 48.73 | 62.15 | 67.83 | 50.33 |
| Patch | 384 | no | -CLS | 576 | - | 85.40 | 196 | 108 | 10 | 5 | 0 | 50.89 | 65.08 | 68.50 | 52.83 |
| Patch | 384 | no | -order | 576 | - | 85.10 | 189 | 111 | 9 | 4 | 0 | 52.16 | 62.92 | 66.67 | 50.67 |
| Combined | 768 | no | -CLS-glb | - | - | 86.43 | 154 | 106 | 13 | 2 | 0 | 47.97 | 57.23 | 69.33 | 56.83 |
| Combined | 768 | no | -glb | - | - | 86.26 | 156 | 107 | 18 | 2 | 0 | 49.24 | 58.77 | 71.00 | 56.17 |
| Combined | 768 | no | -order | - | - | 86.43 | 157 | 103 | 15 | 4 | 0 | 50.25 | 64.46 | 70.17 | 50.00 |
| *RADIO, without cross-attention aggregation* | | | | | | | | | | | | | | | |
| Segment | 384 | no | | - | - | 84.14 | 160 | 78 | 10 | 4 | 0 | 48.22 | 62.46 | 68.33 | 50.50 |
| Cluster | 384 | no | | - | - | 84.36 | 153 | 103 | 13 | 2 | 0 | 50.25 | 63.85 | 67.50 | 52.67 |
| Combined | 384 | no | | - | - | 85.03 | 160 | 86 | 10 | 3 | 0 | 46.57 | 57.38 | 67.00 | 51.33 |
| Segment | 768 | no | | - | - | 86.09 | 147 | 92 | 12 | 4 | 0 | 48.22 | 58.92 | 71.00 | 50.33 |
| Combined | 768 | no | | - | - | 86.55 | 152 | 105 | 12 | 8 | 0 | 47.59 | 59.69 | 70.00 | 52.67 |
| *RADIO + Qwen3-8B Yang et al. (2025a)* | | | | | | | | | | | | | | | |
| Segment | 384 | rms | | 101 | 16.99 | 84.69 | 126 | 69 | 13 | 1 | 0 | 49.11 | 66.62 | 63.33 | 60.50 |
| Cluster | 384 | rms | | 129 | 12.55 | 83.83 | 156 | 95 | 12 | 6 | 0 | 53.81 | 75.54 | 71.33 | 62.83 |
| Combined | 384 | rms | | 138 | 14.72 | 84.88 | 152 | 81 | 13 | 1 | 0 | 50.89 | 75.85 | 69.33 | 64.50 |

Table 6: Detailed evaluation results of various settings, as an addition to Tables 1-5 in the main paper. MMStar detailed scores is not shown here, but can be found in the attachments.

# F ADDITIONAL RESULTS

## F.1 RESULTS ON OTHER LLMS

We additionally conduct a set of experiments on a more recent LLM Qwen3-8B Yang et al. (2025a), with results included in the last few rows of Table 6. The results are consistent with our observation, while having improved CV-Bench and MMstar performance.

## F.2 DETAILED EXPERIMENTAL RESULTS

Table 6 shows the detailed metric scores under various settings (not including MMStar Chen et al. (2024b)). Due to page limits, some of the results are not shown in the main paper. The complete results are included in Table 6.

## F.3 MORE VISUALIZATION

Fig. 7 shows additional patch feature visualization results based on more visual encoders. Fig. 8 shows more visualizations of MLLM's attentions over visual tokens under different settings.

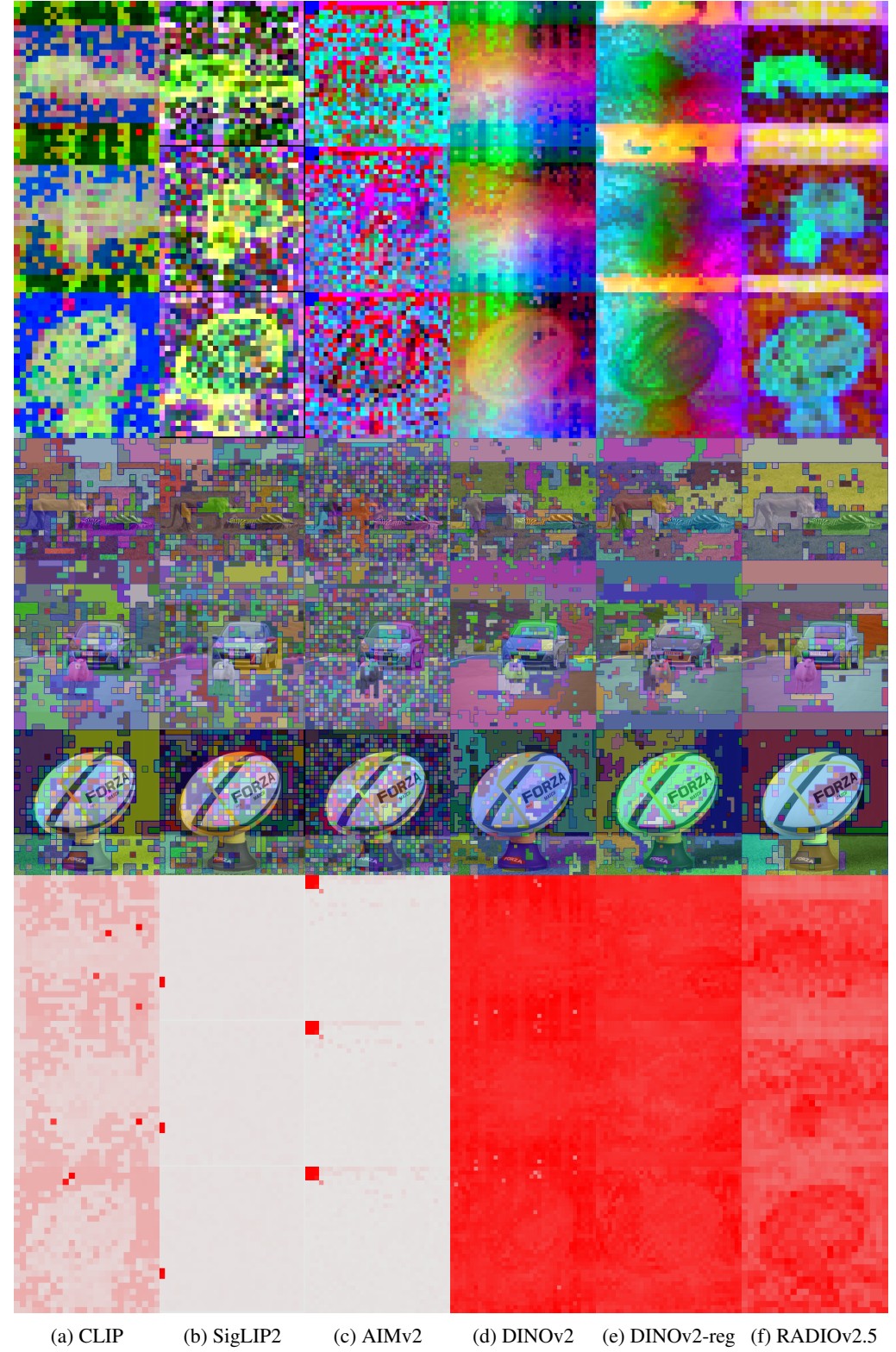

(a) CLIP    (b) SigLIP2    (c) AIMv2    (d) DINOv2    (e) DINOv2-reg    (f) RADIOv2.5

Figure 7: More visualization results for different vision encoders. Rows 1-3: patch feature (PCA); rows 4-6: regions from clustering; rows 7-9: patch feature norm. All results are derived from their native resolution, except for RADIOv2.5, where we use $384 \times 384$. For the visual encoders not mentioned in the main paper, the exact checkpoints used are: AIMv2: `aimv2-large-patch14-448`, DINOv2: `vit_large_patch14_dinov2.lvd142m`, DINOv2-reg: `vit_large_patch14_reg4_dinov2.lvd142m`.

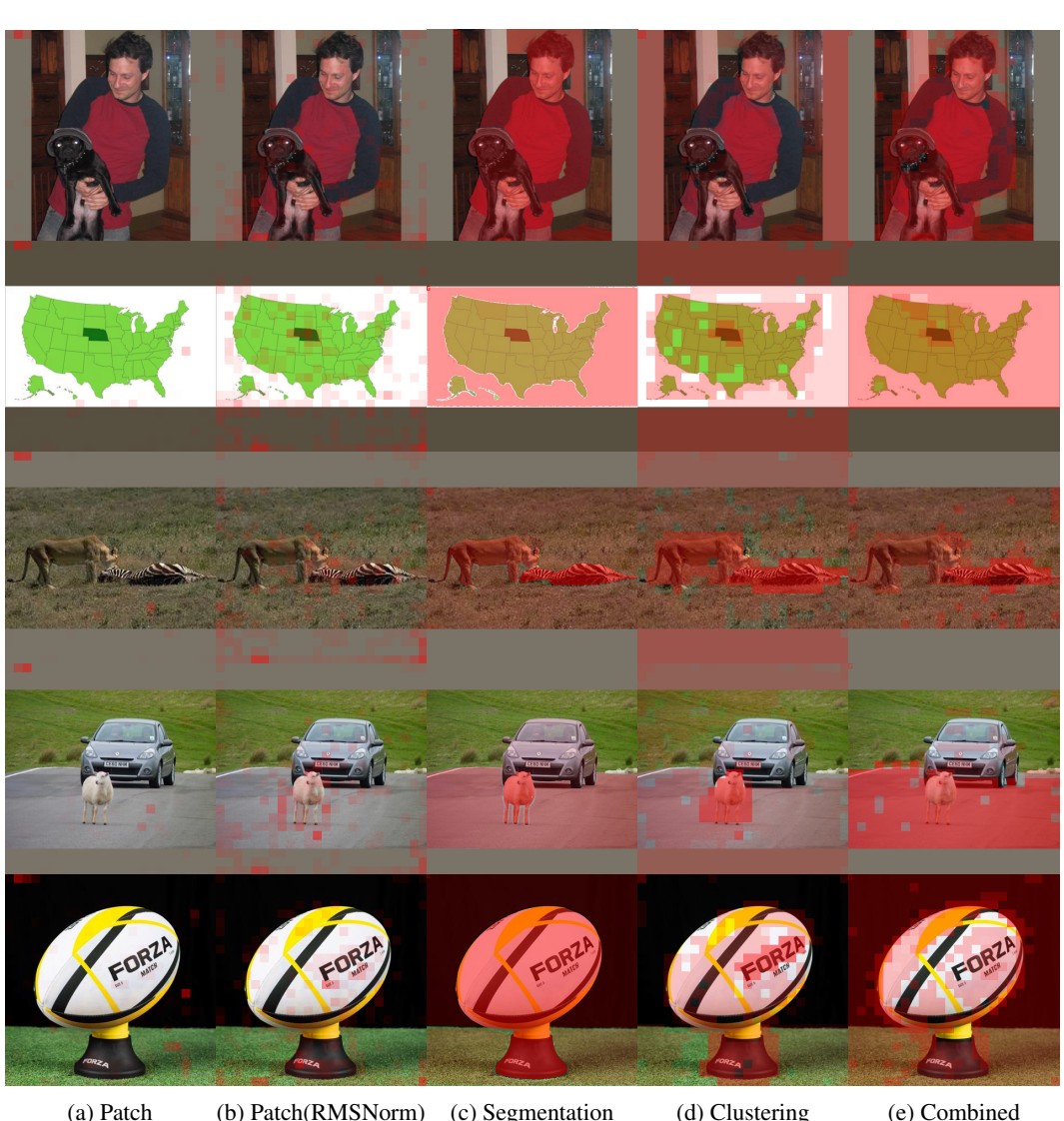

(a) Patch    (b) Patch(RMSNorm)    (c) Segmentation    (d) Clustering    (e) Combined

Figure 8: More visualizations of MLLM's attentions over visual tokens.

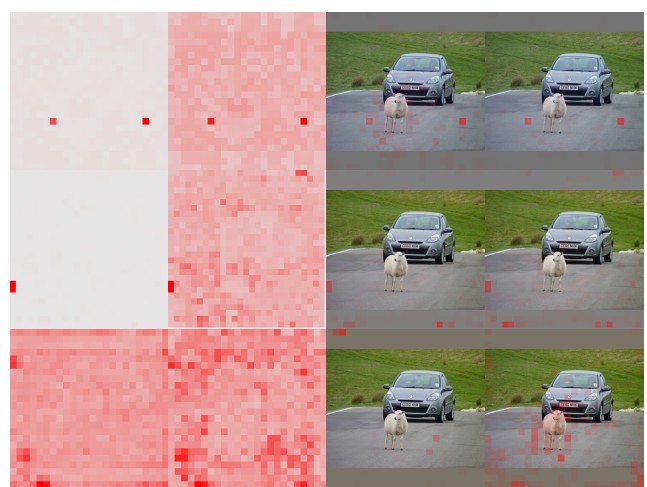

Figure 6: Left: feature norm before/after normalization. Right: attention visualization before/after normalization. From top to bottom: using CLIP, SigLIP2, RADIOv2.5.

