# OpenReview forum: "Dissecting Demystifying Region-Based Representations in MLLMs"
_ICLR.cc/2026/Conference — ICLR 2026 Conference Withdrawn Submission_

### Official Review · Reviewer_8ZZf · 2025-10-30

**Soundness:** 2
**Presentation:** 2
**Contribution:** 3
**Rating:** 2
**Confidence:** 3

**Summary:**

This paper focuses on the effect of aggregating token-level visual representations into region-level visual representations in vision-language models on general multimodal tasks. Authors show how region-level representations improve the performance of MLLM across various multimodal tasks. The authors also conducted a comprehensive discussion and experiments on several related points: (1) how order non-sensitivity of MLLM guarantees that region-level representation won’t collapse the model, (2) how the quality of visual representations influences the effectiveness of region-level representation, (3) how normalization helps region-level representations, (4) how different approaches of forming the region and pooling features influence the effect of region-level representations.

**Strengths:**

The paper transitions from region-based representations’ usage for vision-only tasks to multimodal tasks. The authors ask an interesting question related to “why region-based representations work on MLLM”.  They use an analysis of how the order of the visual token influences the performance of MLLM.

**Weaknesses:**

1. The explanation about the drop in OCR performance needs to be further justified by experiments, e.g., if one creates the region for each independent character, will the performance of region-level representation improve the performance on OCR tasks?
2. The lack of sufficient explanation or label for particular figures makes some of the conclusions less convincing:
    - There is no explanation about the difference between “random order (trained)” and “random order (w/o training)” in Table 3. Does that mean the model is further fine-tuned after reordering the input tokens? Why are these two methods distinguished and compared only in patch-based RADIOv2.5, whereas for CLIP and region-based RADIOv2.5, there is only a single “random order” condition?
    - Why not test the pre-shuffle condition on CLIP and region-based RADIO v2.5?
    - Is there any explanation about why some of the entries are empty in Table 3?
    - Figure 3 does not have any label indicating which image belongs to which model, making it hard to tell anything from the figure here.
3. As discussed in the paper, agglomerative visual encoders could offer better visual representation. To make the conclusion more solid,  I would like to see more results from different agglomerative visual encoders and compare with the traditional visual encoder rather than only RADIOv2.5.
4. The results from Table 4 are not sufficient enough to support the point that RMSNorm helps the region-based representation, as there is only an improvement on MMStar. Also, the format of the plot here is confusing, as there are three models for patch-based representation but only one model for region-based representation. Are these the results for RADIOv2.5? How does RMSNorm work on the region-based CLIP and SigLIP2?

Minor:
1. The location of the “G” letter in figure (b) is different from the “G” in ( c) and (d), also those three “G”s seem to have different luminance.
2. Line 256, the link to the table is not working correctly (Table ??)
3. Figure 6 in the appendix is not centered.

**Questions:**

see weaknesses.

---

### Official Review · Reviewer_Hmy2 · 2025-10-30

**Soundness:** 2
**Presentation:** 2
**Contribution:** 2
**Rating:** 2
**Confidence:** 3

**Summary:**

The paper explores region-based representations as an efficient and interpretable alternative to patch-based representations. It builds on the observation that MLLM performance remains robust to the input order of visual tokens, implying that spatial information is already embedded within patch features. This insight motivates reorganizing patches into semantically coherent regions. The paper also provides comprehensive experimental evaluation and in-depth analysis to support its findings.

**Strengths:**

The paper offers a comprehensive and systematic analysis of region-based representations. It identifies feature smoothness as a key factor underlying their effectiveness and proposes concrete strategies to leverage this property. The visualization and attention analyses are clear and compelling, demonstrating how region-based methods produce more structured and interpretable attention maps while significantly reducing the number of tokens.

**Weaknesses:**

While the paper provides valuable analysis, its methodological novelty is limited. The work primarily examines existing components—such as segmentation, clustering, and normalization—rather than introducing new architectures or learning mechanisms. The performance improvements from region-based representations are moderate, and the exploration of aggregation strategies remains incomplete. In particular, the proposed cross-attention-based aggregation yields marginal gains, indicating that a more sophisticated design may be required. Additionally, the study relies on frozen visual encoders, which restricts insight into how region-based representations might interact with end-to-end optimization or benefit from joint training.

**Questions:**

1. Would unfreezing the visual encoder during fine-tuning enhance feature coherence and potentially reduce the reliance on post-hoc normalization?
2. How sensitive are the results to the number and granularity of regions? Could an adaptive region selection mechanism based on image complexity further improve performance?
3. Is it feasible to integrate the hybrid segmentation–clustering approach into the training process itself, rather than using it solely as a preprocessing step?

---

### Official Review · Reviewer_Y2ak · 2025-10-31

**Soundness:** 2
**Presentation:** 3
**Contribution:** 2
**Rating:** 4
**Confidence:** 3

**Summary:**

This paper study the region-based visual representation for MLLM input instead of traditional patch-based representations that have high computational costs (quadratic token growth) and lack semantic structure
The paper find that MLLM performance is robust to patch token order, as visual encoders encode spatial info into patch features—providing a key basis for region reorganization. It identifies raw visual feature incoherence as the main challenge for region-based representations aggregation . To tackle this, it proposes strategies: using agglomerative backbones (e.g., RADIOv2.5), adding normalization (e.g., RMSNorm), and hybrid regions (SAM segmentation + DBSCAN clustering) .
Experiments show optimized region-based MLLM match patch-based MLLM in performance, while cutting visual tokens for efficiency and boosting interpretability via focused attention .

**Strengths:**

- It is meaningful to investigate how to construct a region-based visual representation suitable for MLLM input—one that can facilitate LLM understanding while reducing training overhead.
- The paper is highly accessible, featuring a well-organized structure that allows readers to easily grasp its core ideas.
- Extensive experiments and visualizations provide solid support for the research findings. The paper conducts in-depth analyses and deduces/validates each finding through systematic reasoning.

**Weaknesses:**

- The paper mainly discusses the region feature aggregation in the vision part. However, how the LLM attends to the patch features and region features remains under exploration.
- The paper proposes a simple way to obtain the region-based representation, which is a post-processing step of the patch vision features. Yet, it does not study how to learn region features (suitable for MLLMs) within the ViT.
- Lack of efficiency discussion: The paper proposes using SAM and clustering methods to extract region-based vision features, but it fails to analyze whether the use of SAM and clustering brings additional computational overhead compared to the patch-based method.
- Missing results in several experiments raise doubts about the correctness of the findings. While Table 1 evaluates 7 benchmarks, Tables 2, 3, and 4 only evaluate on 2 or 4 benchmarks, which may cause confusion for readers.
- Table 3: Configurations E, F, and G are missing results for POPE, OCRBench, and CV-Bench.
- Figure 3 does not indicate which models the visualizations correspond to.
- Typos: Line 256 has a missing reference (marked as `Table ??`).

**Questions:**

- Visualization of the attention mask when altering the order of vision tokens is required. I am curious whether vision tokens in random order exhibit the same positional attention patterns as those in sequential order.
- Many papers on token reduction or token pruning indicate that dropping 75% of tokens even above only slightly impacts performance. Does this mean LLMs do not need to capture all vision tokens and only need to attend to a few vital vision tokens?
- The paper obtain the region-based representation based on SAM mask and token merging. It should compare to other token merging and token pruning methods, like PyramidDrop, (CVPR25), SparseVLM (ICML25), FasterVLM (ICCV25), VisionZip(CVPR2025)
- Each region might be an object or the background with the same concept. Does the region representation for one element correspond to a single token or a set of tokens? Additionally, does the position of regions in the input sequence matter for LLMs?
4. As mentioned in the paper, DINOv2 exhibits better feature coherence due to its self-supervised learning (SSL) training. How about the performance of using the clustering results of DINOv2 features to aggregate CLIP or SigLIP features?
5. In Table 5, what does `768x` denote? If it refers to resolution, why does this setting differ from those in other ablation experiments?

---

### Official Review · Reviewer_xyPG · 2025-11-01

**Soundness:** 2
**Presentation:** 2
**Contribution:** 2
**Rating:** 4
**Confidence:** 3

**Summary:**

This paper aims to dissect the region-based representation in the mul-timodal large language models. The paper shows that region-based representations are robust to patch token order, and their eﬀectiveness depends on smooth, localized visual features. The proposed in-sights are straightforward and easy to follow, though somewhat lacking in depth. The experiments and visualizations are clearly designed to illustrate the ﬁndings, though they sometimes lack quantitative support or deeper analysis.

**Strengths:**

- The insights are straightforward and easy to follow.

**Weaknesses:**

-  Intuitive but Shallow Conclusions: The biggest strength of this paper is also its main weakness. The conclusions are intuitive and easy to follow, but their usefulness for guiding practical applications or informing further theoretical analysis may be limited. They lack underlying theoretical explanations, which makes the paper feel more like a report of experimental observations rather than a thorough dissection.

- Lack of Quantitative Experimental Support: Some of the reasoning would be more convincing if supported by quantitative metrics. For example, in Finding 3, the authors mention that feature non-smoothness poses a challenge for region-based methods. Intuitively, the authors should provide a quantitative analysis of feature non-smoothness and establish its relationship with performance. The absence of such quantitative analyses reduces the depth and the inspirational value of the paper.

**Questions:**

Please refer to the weakness.

---

### Note · Authors · 2025-11-14

I have read and agree with the venue's withdrawal policy on behalf of myself and my co-authors.